# A Mouse-Specific Model to Detect Genes under Selection in Tumors

**DOI:** 10.3390/cancers15215156

**Published:** 2023-10-26

**Authors:** Hai Chen, Jingmin Shu, Carlo C. Maley, Li Liu

**Affiliations:** 1College of Health Solutions, Arizona State University, Phoenix, AZ 85004, USA; hchen294@asu.edu (H.C.); jshu4@asu.edu (J.S.); 2Biodesign Institute, Arizona State University, Tempe, AZ 85281, USA; maley@asu.edu; 3Arizona Cancer Evolution Center, Arizona State University, Tempe, AZ 85281, USA

**Keywords:** cancer genomics, transfer learning, molecular evolution

## Abstract

**Simple Summary:**

We introduce GUST-mouse (Genes Under Selection in Tumors for mouse), a novel computational method designed to identify cancer driver genes within mouse tumor exomes. As the first method of its kind, GUST-mouse transcends conventional frequency-based rules. It incorporates molecular evolutionary theories and leverages transfer learning techniques to effectively differentiate between oncogenes, tumor suppressor genes, and passenger genes. When applied to mouse models of breast cancer, leukemia, and lung cancer, GUST-mouse unveiled that the emergence of somatic driver mutations is profoundly influenced by the genetically engineered background of the mouse models. A comparative analysis with human cancer drivers illuminated both shared and distinct patterns, casting new light on the intricate process of tumorigenesis. The pioneering framework of the GUST-mouse method opens a new avenue for identifying driver genes in non-human cancers.

**Abstract:**

The mouse is a widely used model organism in cancer research. However, no computational methods exist to identify cancer driver genes in mice due to a lack of labeled training data. To address this knowledge gap, we adapted the GUST (Genes Under Selection in Tumors) model, originally trained on human exomes, to mouse exomes via transfer learning. The resulting tool, called GUST-mouse, can estimate long-term and short-term evolutionary selection in mouse tumors, and distinguish between oncogenes, tumor suppressor genes, and passenger genes using high-throughput sequencing data. We applied GUST-mouse to analyze 65 exomes of mouse primary breast cancer models and 17 exomes of mouse leukemia models. Comparing the predictions between cancer types and between human and mouse tumors revealed common and unique driver genes. The GUST-mouse method is available as an open-source R package on github.

## 1. Introduction

Mouse models are indispensable resources that complement human tissues in cancer research [1]. In parallel with large-scale sequencing efforts in human cancers, people have begun sequencing whole exomes and whole genomes of mouse tumors [2,3,4,5]. Sophisticated algorithms have been developed to identify driver genes in human cancers by integrating mutational patterns, somatic evolution, and other informative features extracted from high-throughput sequencing data [6,7,8]. However, such methods are not readily available for non-human organisms. Researchers using mouse tumor models often rely on traditional practices, such as assuming frequently mutated genes as drivers or inferring driver status based on human–mouse orthologs. But not all recurrent mutations are drivers; mutations also occur in hotspots in passenger genes [9,10,11]. Furthermore, because mouse tumor models are often induced with carcinogens or genetically engineered, the accelerated tumorigenesis and limited genetic diversity within the tumors may lead to mutational and selective patterns that are different from those observed in human cancers [4]. Advanced tools are needed to go beyond mutation frequency and sequence homology with human cancer genes to identify bona fide drivers in mice.

Supervised machine learning has been widely used to construct predictive models for cancer driver gene classification, as reviewed in [12]. However, unlike human data, which benefit from meticulously curated driver and passenger gene labels for training supervised models [13], the absence of a well-defined set of labeled genes in mice makes it impractical to train a de novo classifier. To tackle this challenge, we use transductive transfer learning [14], a technique that adapts a classifier trained on labeled data in the source domain (human) to fit unlabeled data in the target domain (mice). Transfer learning is particularly advantageous when the source and target domains share similarities [15]. Given that the fundamental mechanisms of tumor development are largely conserved in human and mouse tumors [16], predictive models built using human genes may be leveraged to develop models for mice.

We previously developed the GUST (Genes Under Selection in Tumors) method that distinguishes between oncogenes (OGs), tumor suppressor genes (TSGs), and passenger genes (PGs) in human cancer genomes [7]. GUST has two functionalities. Firstly, it estimates the critical evolutionary parameters that characterize the long-term conservation of genetic sequences based on multiple sequence alignment among species [17,18] and short-term fitness selection based on somatic mutations in tumor exomes [19,20]. This functionality does not require model training and the algorithms can be directly applied to mouse genes. Secondly, GUST includes a random forest classifier that predicts cancer driver genes using the evolutionary parameters and mutation distribution features. We developed this functionality for mice via transductive transfer learning. To adapt a random forest model, we employed structure reduction that progressively pruned the trees [21], and threshold shifting that adjusted the cutoff value used at each node split [22]. We then applied the new GUST-mouse method to mouse exomes of genetically engineered breast cancer models and leukemia models. Comparing the predictions between cancer types and between human and mouse tumors revealed common and unique driver genes.

## 2. Materials and Methods

We summarized the workflow of GUST-mouse development and application in schematic representations (Figure 1).

### 2.1. Source and Target Domains

The random forest model in the GUST method is trained to classify human genes into OGs, TSGs, and PGs. The source domain for this model consists of 533 labeled human genes in various cancer types, referred to as “hBenchmark”. For each gene, position-specific evolutionary rates, which represent long-term conservation, were computed using the Multiz alignments of protein sequences from 100 vertebrates [23]. Short-term somatic selection and mutational distribution features were derived using tumor exomes from the Cancer Genome Atlas Program (TCGA) [24]. These data were part of the published Appendix A of GUST. The human reference genome used was GRCh38 (hg38).

The target domain was mutated genes in mouse tumors. In one application, we used 65 mouse exomes of primary breast cancers (mmBRCA) from a published study of mouse models of breast cancer [25]. In another application, we used 17 mouse exomes of leukemia (mmLKM). The VCF files containing somatic mutations in each sample are available from the NCBI GEO database (GSE142387 for breast cancer and GSE137542 for leukemia). The mouse reference genome used was GRCm38 (mm10).

### 2.2. Estimating Parameters of Long-Term Species Evolution for Mouse Genes

Since the Multiz alignments use humans as the reference species, we swapped the mouse sequence with the human sequence, removed sites where the mouse sequence contained a gap, and verified that the resulting mouse sequences were identical to those in the mm10 genome. The evolutionary rate (r) at each position was computed using the Fitch method [26], expressed as the number of substitutions per billion years (s/bys). For a given gene, we computed the mean evolutionary rate over all mutated positions (denoted as E.gene) and the evolutionary rate of the most frequently mutated position (E.summit).

### 2.3. Estimating Parameters of Short-Term Somatic Evolution in Mouse Tumors

The algorithm to estimate somatic selection on a mutated gene is described in the GUST method [7]. Briefly, for each protein-coding gene, saturated point mutations are simulated to infer the expected mutational patterns, considering factors such as codon usage, mutation types, and varying mutational rates. Synonymous mutations are used as the neutral baseline. When analyzing a gene that is mutated in a set of mouse tumors, the observed mutation pattern is compared with the expected pattern to infer selection coefficients of missense mutations (ω) and protein-truncating (nonsense and frameshifting indel) mutations (φ) using maximum likelihood estimation. After log transformation, the sign of a selection coefficient indicates the direction of selection, and the value indicates the magnitude. A coefficient close to zero indicates neutral selection.

### 2.4. Extracting Features Describing Mutation Distribution

The GUST program captures the mutational profile of a gene using several features including fractions of missense mutations and protein-truncating mutations (denoted as R.missense and R.truncating, respectively), sizes of clusters of mutations forming hotspots (R.peak, R.summit, and C.summit), and lengths of truncated peptides (R.length). The GUST program’s functions for calculating these features can be used to analyze and characterize mutational patterns in mouse genes from exome sequencing data.

### 2.5. Refining the Random Forest Classifier

The random forest model in GUST uses 10 predictors, including two long-term evolutionary parameters (E.gene and E.summit), two short-term evolutionary parameters (log(ω) and log(φ)), and six mutational distribution parameters (R.missense, R.trunating, R.peak, R.summit, C.summit, and R.length), to classify genes into OGs, TSGs, and PGs. For a given mouse gene with mutations observed in a set of cancer exomes, GUST-mouse calculates the values of these predictors. Although the class labels of mouse genes are unknown, it is reasonable to assume that genes belonging to the same class tend to cluster together based on these predictors. GUST-mouse then calculates pairwise Euclidean distances (*D*) between mouse genes and examines how these distances change in each node of the tree. This allows GUST-mouse to refine the classifier based on the patterns of similarity or dissimilarity between genes in the tree nodes, even in the absence of known class labels for the mouse genes.

Given a bifurcating decision tree *T_h_* in the random forest classifier *RF* trained on human data, GUST-mouse implements two types of transductive transfer learning. The first type involves pruning the tree via structure reduction [21]. Specifically, it traverses the *T_h_* tree from root to leaves in a depth-first order. At each internal node, GUST-mouse calculates the mean distance between all pairs of genes reaching that node (*D_i_*), as well as between all pairs of genes reaching each of its child nodes (*D_a_* and *D_b_*). If splitting the internal node into the child nodes does not reduce the pairwise gene distance (i.e., *D_i_* < *D_a_* and *D_i_* < *D_b_*), the clade below the internal node is snipped. This process is applied recursively to the entire tree, resulting in an updated tree *T_prune_*.

The second type of transductive transfer learning in GUST-mouse does not change the topology of the tree, but rather adjusts the splitting threshold of each internal node [22]. Similar to the first type, GUST-mouse traverses the *T_h_* tree from root to leaves in a depth-first order. At each internal node, the optimal threshold of the splitting feature is selected to minimize the sum of pairwise gene distance in the two child nodes (i.e., *argmin_t_ (D_i_ + D_a_)* where *t* is the splitting threshold). After completing the traversal and threshold adjustment, the updated tree *T_shift_* is obtained.

The GUST model consists of 200 *T_h_* trees. By applying structure reduction and threshold adjustment to each tree, the GUST-mouse model will have 200 *T_prune_* trees and 200 *T_shift_* trees. These updated trees collectively constitute the random forest classifier *RF-mouse*. It is important to note that the original *RF* model and the subsequent transferred *RF-mouse* model were both trained to learn the general patterns distinguishing between OGs, TSGs, and PGs. When applying the model to make predictions, users may choose to use tumor exomes of the same cancer type or different types to predict drivers in specific cancer type or pan-cancer drivers.

### 2.6. Evaluation of GUST-Mouse Performance

The scarcity of curated murine cancer-related genes hinders the conventional verification of GUST-mouse predictions. To address this issue, we matched mouse genes to human orthologs and used the classification labels of human orthologs as surrogates of ground truth. We also compared GUST-mouse predictions with those using the classical 20/20 rule [27], which classified genes with >20% truncating mutations as TSGs and genes with >20% missense mutations at recurrent positions as OGs. We examined the concordant and discordant rates between these two methods.

## 3. Results

### 3.1. Human Genes and Mouse Genes Showed Similar Distributions of Evolutionary Parameters

For each mutated gene in the hBenchmark, mmBRCA, and mmLKM datasets, we calculated the evolutionary rate of each affected position. Low evolutionary rates indicate strong purifying selection across species and strong functional impact. The comparison of evolutionary rates between the human and mouse mutated genes showed highly similar distributions (Figure 2A). This suggests that the evolutionary constraints and functional impact of mutations are comparable between human and mouse tumors, despite the species’ divergence.

The selection coefficients, log(ω) and log(φ), quantify short-term somatic selection on missense mutations and protein-truncating mutations, respectively. The sign of the coefficients indicates the direction of selection (positive or negative) and the magnitude indicates the strength of selection. We computed these values for mutated genes in the hBenchmark, mmBRCA, and mmLKM datasets. The scatterplots showed that the distribution of log(ω) and log(φ) in the human and mouse datasets shared a similar pattern—genes under neutral selection (values close to 0) were clustered separately from genes under directional selection (values deviated from 0, Figure 2B).

The consistent patterns of long-term and short-term evolutionary parameters across datasets confirmed that human and mouse cancers share common mechanisms. Therefore, findings from one species can be informative and relevant for understanding cancer biology in the other species, which justifies the use of transfer learning approaches.

### 3.2. Unsupervised Euclidean Distance Was a Good Proxy of Supervised Splitting Index

The *RF* model contained 200 *T_h_* trees trained on the labeled hBenchmark data representing the source domain. We previously reported that this model had a cross-validation accuracy of 92% and area under the receiver operating characteristic curve (AUROC) of 0.97 [7]. In the training of the *RF* model, the Gini impurity score was used as the splitting index. To assess whether within-node Euclidean distance calculated without knowing class labels was a good proxy of Gini impurity score, we examined each split where a parent node was divided into two child nodes. In 99.6% (1800 out of 1807) of the splits, the mean pairwise distance of genes in a child node was smaller than that in the parent node. This observation is consistent with the expectation that node splitting creates clusters of similar genes. Furthermore, the within-node distance was positively correlated with the Gini impurity, and the correlation was stronger in large nodes close to the root than in small nodes close to the leaves (Pearson correlation coefficient range from 0.88 to 0.31, all *p* < 10^−16^, Figure 2C,D). This result confirmed our assumption that genes with different class labels form clusters that can be inferred from Euclidean distance.

### 3.3. Adapted Random Forest Classifier Predicted Driver Genes in Mouse Tumors

The mmBRCA dataset contained primary breast cancer tumors from 53 MMTV-PyMT transgenic mice that modeled PI3K activation [28] and 12 MMTV-Her2 transgenic mice that modeled *Her2* overexpression [29]. The mmLKM dataset contained 17 leukemia thymus samples from NP23-NHD13 double transgenic mice that modeled inhibited hematopoietic differentiation [30]. Because GUST required at least five protein-coding mutations in each gene, we removed genes with few mutations. This filtering produced 218 genes containing 2479 mutations in the mmBRCA(PyMT) tumors, 107 genes containing 1698 mutations in the mmBRCA(Her2) tumors, and 27 genes containing 262 mutations in the mmLKM tumors. Using these data as the target domain, we adapted the human *RF* model to build the RF-mouse model. We then applied the RF-mouse model to predict driver genes in each of the three sets of tumors. Using the predictive probability >0.5 cutoff, we identified 46 Ogs and 16 TSGs in mmBRCA(PyMT) tumors, 26 Ogs and 4 TSGs in mmBRCA(Her2) tumors, and 16 Ogs and 1 TSG in mmLKM tumors (Table 1, Appendix A).

The *Ddx42* gene in mmLKM tumors was predicted as an OG with the highest probability (0.995). As expected, missense mutations in this gene were clustered in a hotspot under strong positive selection (log(ω) = 5.0), while truncating mutations were completely missing and under strong negative selection (log(φ) = 5.0, Figure 3A). *Ddx42* is a member of the DEAD/H-box helicase family that is broadly classified as oncogenes in various cancers, including leukemia [31]. Although somatic mutations in human *DDX42* are infrequent, overexpression and copy number gain of this gene have been reported in blood cancers, supporting its oncogenic roles [32,33].

The *Foxn2* gene in mmBRCA(PyMT) tumors is a representative example of TSGs. Nonsense mutations (S108*, P114*, and Y115*) that truncated the protein and removed the DNA-binding domain were under strong positive selection (log(φ) = 3.54, Figure 3B). A previous study reported that the human ortholog *FOXN2* gene was significantly downregulated in breast cancer tissues and cell lines [34]. Further in vitro experiments confirmed that ectopic expression of *FOXN2* suppressed the proliferation of breast cancer cells, and the inhibition of *FOXN2* promoted tumor growth, strongly supporting the tumor suppressor roles.

An overwhelming majority of the mutations were predicted as PGs, which interestingly included *Her2* (also known as *Erbb2*), a well-known OG in breast cancer. As previously reported [35], *Her2* was frequently mutated in MMTV-Her2 mouse models. In the mmBRCA(Her2) dataset, *Her2* was mutated in 12 out of the 17 tumors, with a median of 18 mutations per tumor (range 5 to 31). A total of 99 synonymous mutations, 95 missense mutations, and 16 truncating mutations were scattered across the length of the gene without forming any hotspots. GUST-mouse estimated that the missense mutations and truncating mutations were both under neutral selection (log(ω) = –0.78, log(φ) = 0.26, Figure 3C). This pattern is drastically different from that in TCGA-BRCA human breast cancers, where *HER2* mutations formed a prominent hotspot around amino acid position 755 to 777 within the drug-targeted kinase domain (Figure 3D) [36]. These results imply that in mmBRCA(Her2) mice, where *Her2* is genetically engineered to constitutively overexpress, additional activating mutations in the original copy of this gene may not confer significant growth advantages to tumor cells compared to functionally neutral mutations.

### 3.4. Comparison between Cancer Types Revealed Common and Unique Drivers

We first compared the two sets of mouse breast cancer samples to examine how genetically engineered germline exposure influences the somatic selection. The mmBRCA(PyMT) and mmBRCA(Her2) tumors shared 51 commonly mutated genes, each harboring at least five somatic mutations. GUST-mouse made concordant predictions for 45 of these genes (88.2%) in the two datasets, including 39 PGs, four OGs, and two TSGs (Figure 4A). For example, the *Chuk* gene showed highly similar mutational patterns in the two tumor sets and GUST-mouse concordantly predicted it as a TSG (Figure 4B,C). *Chuk* is involved in mammary gland development and has been proposed as an emerging tumor suppressor in several organs of humans and mice [37]. Other common drivers include *AY358078*, *Lama2*, *Pcdhb18*, and *Zfp982* as OGs and *Vmn2r72* as a TSG.

Only six genes received different predictions between the two tumor sets, and the classifications always switched between driver and passenger roles. Among these cases, two genes were predicted as TSGs in one tumor set but as PGs in the other tumor set. Likewise, four genes changed from OGs to PGs. For example, *Naip2* and *Naip5* are anti-apoptosis genes and were predicted as OGs in the mmBRCA(Her2) tumors but as PGs in the mmBRCA(PyMT) tumors. In mmBRCA(Her2) tumors, missense mutations in these two genes were clustered into hotspots under positive selection (Figure 4D,E). The predicted OG roles align with their apoptosis inhibitory functions and overexpression in human breast cancers [38]. However, these two genes harbored only a few missense and synonymous mutations in the mmBRCA(PyMT) tumors and were predicted as PGs (Figure 4F,G). We found no dual-role genes that switched class between OG and TSG.

We then compared mouse breast cancers with leukemia. Only two genes, *Naip5* and *Eef2*, harbored at least five mutations in both mmBRCA and mmLKM tumors. As mentioned above, the classification of *Naip5* varied in different tumor contexts, being an OG (probability = 0.73) in the mmBRCA(Her2) tumors but a PG in mmBRCA(PyMT) and mmLKM tumors. The *Eef2* gene encodes a highly conserved eukaryotic translation elongation factor essential for protein synthesis. It was predicted as an OG in both mmBRCA(PyMT) tumors and mmLKM tumors (probability = 0.77 and 0.85, respectively; Figure 4H,I). In these tumors, the missense mutations in *Eef2* were clustered at amino acid positions 83 to 88. Overexpression and amplification of *EEF2* protein have been reported in various types of human cancers, including breast cancer and leukemia [39,40]. By regulating cell death, it promotes tumor cell proliferation and correlates with poor prognosis of several types of cancers [41,42]. These results suggest *Eef2* as a pan-cancer OG.

### 3.5. Human–Mouse Comparisons

Due to the lack of known driver genes in mice, we were unable to directly assess the accuracy of GUST-mouse predictions. To address this issue, we took a creative approach—matching mouse genes to human orthologs and using the classification labels of human orthologs as surrogates of ground truth. Initially, we attempted to use 533 human genes in the GUST training data hBenchmark. Unfortunately, none of these human genes harbored at least five mutations in the mouse tumors; thus, GUST-mouse could not make predictions. In our second attempt, we predicted drivers in human breast cancer by applying GUST to the TCGA-BRCA samples and then compared these to GUST-mouse predictions in the mmBRCA samples.

We found 45 human–mouse orthologs that were mutated in both the TCGA-BRCA samples and the mmBRCA samples. For the human genes in these ortholog pairs, GUST predicted three TSGs, 42 PGs, and no OGs. For the corresponding mouse genes, GUST-mouse predicted two TSGs, seven OGs, and 36 PGs (Appendix A). The predictions were concordant for 35 human–mouse orthologs (77.8%), including 43 pairs of PGs and 1 pair of TSGs (Figure 5A). The common TSG ortholog was the human *PTPDC1* and mouse *Ptpdc1*. Interestingly, even though the sample size of the mouse tumors (65) was much smaller than that of the human tumors (952), more missense and truncating mutations were observed in mouse tumors (Figure 5B,C), leading to higher prediction probability (0.89 vs. 0.61).

Among the 10 ortholog pairs receiving discrepant classifications, 7 were mouse OGs predicted as PGs in human tumors (Appendix A). One of them was the above-mentioned *Eef2* gene that showed the signature mutational pattern of OG in mmBRCA(PyMT) tumors (Figure 4H). Conversely, missense mutations in the human ortholog *EEF2* were scattered, not forming any hotspots, and were only under weak positive selection (log(ω) = 1.37, Figure 5D). The current literature supports the notion that *EEF2* is an OG in human breast cancer due to gene duplication and overexpression rather than point mutations and short indels [39,40]. Thus, it is plausible that this gene functions as an OG in both human and mouse tumors, but through different genetic alterations. Similarly, for three additional putative mouse OGs, existing studies indicate that their human orthologs promote breast cancer growth via gene overexpression, copy number gain, or epigenetic activation, while point mutations are rare. These human–mouse orthologs included *CFH-Cfh4* [43,44], *GLUL-Glul* [39,40], and *ITGAD-Itgad* [45,46]. Confirming these four orthologs as OGs increased the number of human–mouse genes with consistent roles to 39 (86.7%). The remaining three putative mouse OGs, namely *Lama2*, *Pcdhb18*, and *Ppig*, lacked evidence to support the OG role of their human orthologs. This discrepancy may suggest differences between species or false positive predictions.

Assuming that human–mouse orthologs have the same functions in breast cancer tumorigenesis and the predictions of human driver genes are reliable, we estimated the performance of GUST-mouse. The overall accuracy was 86.7% (39 out of 45 genes receiving concordant classifications). The positive predictive value was 57.1% for OG predictions (four out of seven genes) and 50% for TSG predictions (one out of two genes). The negative predictive value was 94.4% (34 out of 36 genes). The sensitivity was 33.3% for TSGs (one out of three genes) and 100% for OGs (four out of four genes). The specificity was 89.5% (34 out of 38 genes). However, it is important to note that when the assumptions are violated in practice, the real performance of GUST-mouse might deviate from the estimated values.

The “gene gold age paradox” posits that simpler ancestors of more complex organisms possessed better optimized genes [47]. However, this paradox cannot be applied universally. Despite purifying selection being generally weaker in primates compared to rodents, it is stronger in genes involved in development and expression regulation in primates. In contrast, in rodents, it favors genes associated with metabolism, transport, and energetics. To investigate whether such differences also existed among cancer driver genes, we extracted and analyzed the d/m ratios from the human–mouse–cattle triad, which estimated the strength of purifying selection [47]. We found that the d/m ratio of cancer driver genes in human breast cancer was significantly lower than that of passenger genes (mean = 0.068 vs. 0.175, *t*-test *p* = 0.04), implying stronger purifying selection in humans. This result aligns with our expectations, as cancer-related genes are enriched in pathways involving gene expression regulation, cell signaling, and stem cell proliferation and differentiation. However, in mouse breast cancers, the difference in the d/m ratios between driver and passenger genes was not statistically significant (*p* = 0.74), likely due to the small sample size.

### 3.6. Comparison with the 20/20 Rule

The 20/20 rule, a frequency-based guideline that can be applied across species, classifies genes with >20% truncating mutations as TSGs and genes with >20% missense mutations at recurrent positions as OGs [27]. We applied this rule to the mmBRCA and mmLKM data and compared the results with the GUST-mouse predictions (Appendix A). We found that all OGs (88 of 88) and 71.4% TSGs (15 of 21) identified by the GUST-mouse model were also consistent with the 20/20 rule. For two of the TSGs (*Chuk* and *Nme2*, Appendix A) that were identified using the GUST-mouse model but not using the 20/20 rule, substantial existing experimental evidence supported their tumor-suppressing activities [48,49]. In contrast, the 20/20 rule classified 131 OGs that were not identified using the GUST-mouse model. For 86.3% (113) of these genes, missense mutations were not under positive selection (selection coefficient < 1), indicating that they are likely not genuine OGs. The 20/20 rule also classified the two additional TSGs (Appendix A); while the *Siah1a* gene is a known TSG [50], none of the current literature associates the *Mrho2a* gene with cancer. These comparative analyses suggest that the GUST-mouse model aligns well with the 20/20 rule in identifying driver genes but exhibits a much lower false positive rate.

### 3.7. Classifying Driver Genes in Tumors with Low Mutation Rates

The transfer learning techniques require a reasonably sized unlabeled dataset in the target domain. For tumors with only a small number of mutant genes, it is not possible to adapt the GUST model to fit that specific dataset. To examine whether the GUST-mouse model adapted to the mmBRCA and mmLKM data can be used to make predictions in other cancer types, we analyzed the exomes of a set of primary lung adenocarcinoma in mice [51]. These mice were genetically engineered to harbor mutant *Egfr*, mutant *Kras*, or the overexpression of *Myc* oncogenes. Due to a low mutation rate, only two genes (*Kras* and *Rrs1*) exhibited at least five somatic mutations and could be analyzed using GUST-mouse. The GUST-mouse model predicted *Kras* as an oncogene and *Rrs1* as a passenger gene, consistent with their established functional roles (Appendix A). As expected, the *Kras* somatic mutations were observed in tumors driven by the *Egfr* mutant or *Myc* overexpression, but not in tumors driven by the genetically engineered *Kras* mutant. This result supports the use of the pre-trained GUST-mouse model to predict driver genes in other cancer types.

## 4. Discussion

Tumorigenesis is an evolutionary process, in which selectively advantageous mutations accumulate in cancer cells, leading to uncontrolled cell growth and tumor formation [8,52]. The newly developed computational tool, GUST-mouse, is the first of its kind to enable the study of mouse tumors within an evolutionary framework. It provides two levels of analysis—estimation of evolutionary parameters and classification of driver genes.

From a long-term evolutionary perspective, cancer driver mutations are under strong purifying selection across species and tend to affect highly conserved sites in the genome [53]. GUST-mouse, through its ability to compute substitution rates at positions affected by different types of somatic mutations, provides valuable information about the evolutionary conservation of mutated sites. Similarly, from a short-term evolutionary perspective, mutations that result in gain-of-function or loss-of-function effects are under strong directional selection, as measured via the selection coefficients (ω and φ). These quantitative measures can assist researchers in biomarker selection and understanding the molecular mechanisms underlying cancer development. Other computational methods can also incorporate these values as prior knowledge into algorithm design.

Due to the scarcity of curated cancer drivers in mice, GUST-mouse relies on transfer learning to adapt the classifier trained on labeled human data to fit in the mouse domain. An important consideration in transfer learning is the similarity between the source domain and the target domain. Theoretically, tumorigenesis in humans and tumorigenesis in mice share common hallmarks [16]. Our empirical analysis confirmed that the human and mouse exome data indeed shared similar distributions (Figure 2). Using this adapted classifier, we identified known cancer drivers and passengers, with patterns consistent with expectations (Figure 3). However, models constructed from unlabeled data may have intrinsic weaknesses. While the current literature reports that structure reduction and threshold shifting are effective techniques to transfer a random forest model, we were unable to directly evaluate the performance of the GUST-mouse classifier. As an alternative, we compared putative mouse driver genes with human driver genes and estimated the lower bounds of GUST-mouse accuracy. To provide transparency to users, GUST-mouse displays a warning message of “accuracy unknown” in the header of the prediction result file. This serves as an alert to users to interpret the results with caution, considering the potential uncertainties. Further research and validation using labeled data in the target domain may be necessary to assess and improve the performance of the GUST-mouse classifier.

Different mouse models may undergo distinct interventions and may use different sequencing technologies. A robust classifier shall accommodate the extensive heterogeneity inherent in the data. Because the GUST-mouse model reported in the manuscript was adapted to mouse breast cancer and leukemia data, it is most suited to make predictions in similar cancer types. For other cancer types, we suggest adapting the human GUST model to the specific data, provided that there is a substantial mutation count (hundreds of genes, each harboring at least five coding mutations). However, in scenarios with limited mutated genes, using a GUST-mouse model trained on different cancer types is a feasible alternative, as the model is expected to capture the evolutionary and mutational patterns of driver and passenger genes in general. Indeed, the applicability of this approach is corroborated by the analysis of the mouse lung cancer data. The low mutation rate in the mouse lung cancer models precluded the training of a new GUST-mouse model. Despite this limitation, the existing GUST-mouse model yielded predictions congruent with the established gene functions, affirming the model’s versatility.

The human–mouse comparisons revealed that an overwhelming majority of the predictions were concordant for human–mouse orthologs. However, exceptions existed, such as the *HER2* gene that is a well-known OG in human breast cancer, but its mouse ortholog harbored abundant functionally neutral somatic mutations. These exceptions highlighted the diverse landscape of tumor genomes, in which somatic mutation is only one of several types of genetic alternations that may affect the activities of a gene. As GUST-mouse classifies genes solely based on somatic mutations, the predictions shall be combined with additional independent evidence to interpret the overall functional impact of a gene. It is also noteworthy that genetically engineered mouse models produced tumors in a much shorter time than human models (months vs. decades) [25]. Therefore, we would expect fewer drivers with stronger impact in these tumors.

In addition to mice, cancer studies in other species such as dogs are steadily increasing [54]. We expect that the transfer learning approach employed in GUST-mouse will be suited to construct predictive models to classify driver genes in these datasets as well. Animal models, due to their genetically engineered backgrounds and inherited differences from human models, may only partially manifest the tumorigenesis process in human cancers. Assuming a similar driver status for orthologous genes is an oversimplification. The GUST-mouse method represents the first effort to use sophisticated computational methods to predict driver genes in non-human tumors. We implemented GUST-mouse as an R package that is freely available on github (https://github.com/liliulab/gust.mouse). Detailed documentation is provided in the standard R manual format.

## 5. Conclusions

The GUST-mouse method provides a mouse-specific model to study the long-term and short-term evolution of cancer mutations, and to identify driver genes. It is a valuable computational tool that can contribute to our understanding of tumorigenesis and facilitate comparative studies between human and mouse tumors.

## Figures and Tables

**Figure 1 cancers-15-05156-f001:**
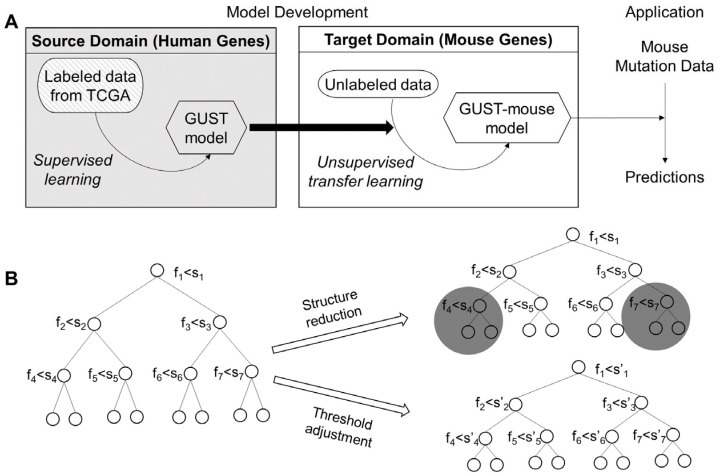
Schema of the GUST-mouse model development. (**A**) The human GUST model trained on labeled data (source domain) was adapted to fit the unlabeled mouse data (target domain). The derived GUST-mouse model was applied to mouse exome mutation data to make predictions. (**B**) The human GUST model is a random forest classifier consisting of 200 decision trees. For each decision tree in the forest, structure reduction was applied to prune branches (dark circles) and threshold adjustment was applied to update splitting cutoff values (s to s’ at each node), which produced two new decision trees.

**Figure 2 cancers-15-05156-f002:**
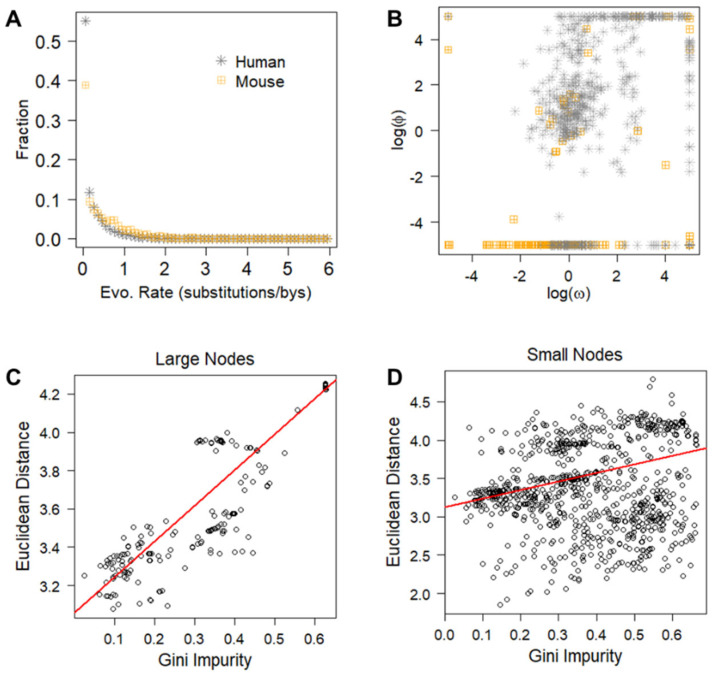
Building the RF-mouse classifier via transfer learning. (**A**) Mutations in the human dataset (hBenchmark) and the mouse dataset (mmBRCA + mmLKM) showed similar distributions of long-term evolutionary rates. (**B**) Scatterplots of short-term somatic selection of missense mutations measured using log(ω) and truncating mutations measured using log(φ) showed similar distributions in the human dataset and the mouse dataset. (**C**,**D**) Gini impurity score and within-node Euclidean distance were strongly correlated in large nodes containing >200 genes (**C**) and were moderately correlated in small nodes containing <20 genes (**D**). Red lines represent linear fits.

**Figure 3 cancers-15-05156-f003:**
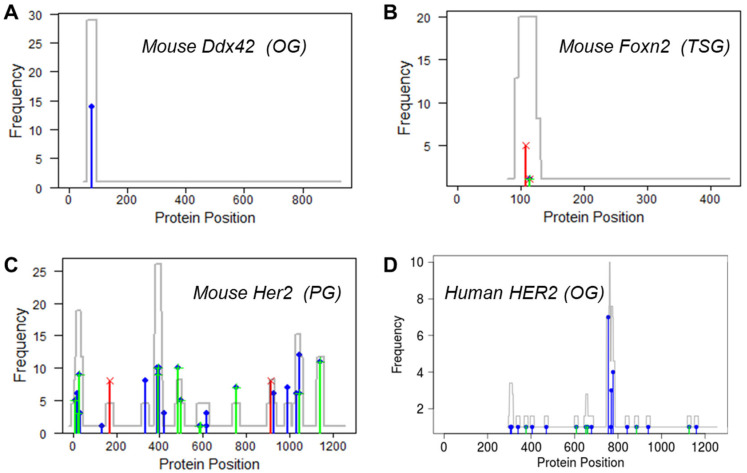
Mutation profiles of representative genes. (**A**) The *Ddx42* gene was predicted as an OG, showing a hotspot of missense mutations (blue bars). (**B**) The *Foxn12* gene was predicted as a TSG, showing a cluster of protein-truncating mutations (red bars). (**C**) The *Her2* gene was predicted as a PG, with synonymous mutations (green bars), missense mutations, and truncating mutations scattered throughout the protein. (**D**) Human *HER2* gene is an OG, showing a hotspot of missense mutations. Gray lines are mutation density plots.

**Figure 4 cancers-15-05156-f004:**
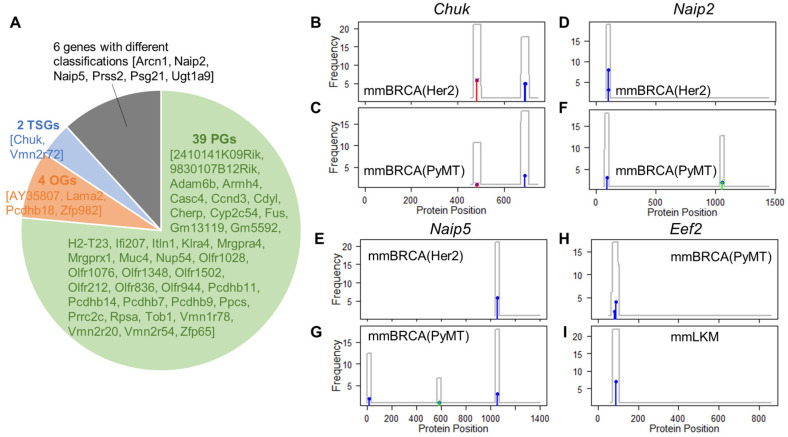
Comparison between cancer types. (**A**) Pie chart shows genes receiving the same or different classifications between the mmBRCA(PyMT) tumors and mmBRCA(Her2) tumors. Gene names are displayed inside square brackets. (**B**,**C**) The *Chuk* gene was concordantly predicted as a TSG. (**D**–**G**) The *Naip2* and *Naip5* genes were predicted as OGs in mmBRCA(Her2) tumors but as PGs in mmBRCA(PyMT) tumors. (**H**,**I**) The *Eef2* gene was predicted as OGs in both mmBRCA(PyMT) tumors and in mmLKM tumors. Frequencies of truncating mutations (red bars), missense mutations (blue bars), and synonymous mutations (green bars) at each protein position are plotted. The gray line represents mutation density.

**Figure 5 cancers-15-05156-f005:**
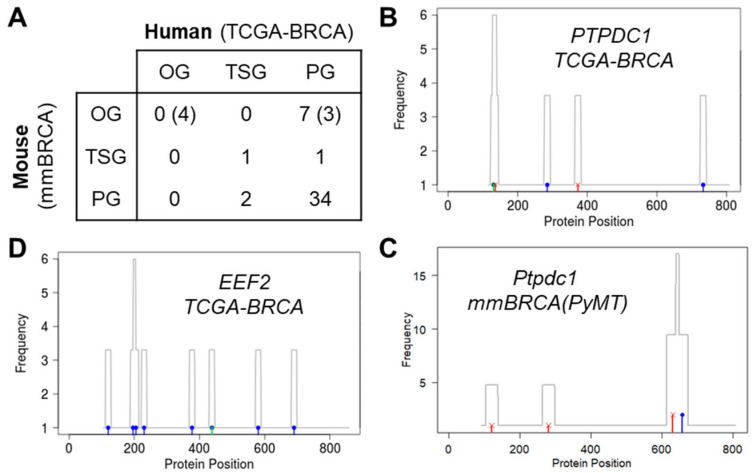
Comparison of human–mouse orthologs in breast cancers. (**A**) Classification of 45 pairs of human–mouse orthologs. Values inside parentheses are after adjustments of the 4 OGs. (**B**–**D**) Distribution of missense, nonsense, and synonymous mutations (blue, red, and green bars, respectively) in human *PTPDC1* gene (**B**), mouse *Ptpdc1* gene (**C**), and human *EEF2* gene (**D**). Gray lines are mutation density plots.

**Table 1 cancers-15-05156-t001:** Ogs and TSGs predicted with high confidence (probability > 0.8).

	Symbol	log(ω)	log(φ)	Class	Prob.		Symbol	log(ω)	log(φ)	Class	Prob.
mmLKM	*Ddx42*	5	−5	OG	0.995	mmBRCA(PyMT)	*Macroh2a2*	5	−5	OG	0.992
	*Zcchc4*	5	−5	OG	0.995		*Ybx1*	5	−5	OG	0.985
	*Riox2*	5	−4.99	OG	0.99		*Fbxo5*	5	−5	OG	0.98
	*Trav7-6*	5	−4.99	OG	0.98		*Nudt15*	5	−4.61	OG	0.975
	*Ubqln5*	5	−4.6	OG	0.98		*Fam205c*	5	−5	OG	0.97
	*Cyp2u1*	5	−5	OG	0.965		*Cfhr4*	5	−5	OG	0.97
	*Pram1*	5	−5	OG	0.952		*Sbpl*	5	−5	OG	0.97
	*Tbc1d2b*	5	−5	OG	0.952		*Marchf5*	5	−5	OG	0.967
	*Dpagt1*	5	−5	OG	0.937		*Itgad*	5	−5	OG	0.965
	*Eef2*	5	−5	OG	0.847		*Glul*	5	−5	OG	0.957
	*Ankrd13a*	5	−5	OG	0.839		*Kng2*	5	−5	OG	0.942
	*Gcgr*	5	−5	OG	0.819		*Zfp970*	5	−5	OG	0.879
	*Sult2a6*	5	−5	OG	0.819		*Psg21*	5	−5	OG	0.847
	*Zfp987*	5	−5	OG	0.819		*Trim43b*	5	−5	OG	0.847
	*Prl2c1*	−5	5	TSG	0.862		*Sh2d1b1*	5	−4.99	OG	0.847
mmBRCA(Her2)	*H2-K1*	5	−4.98	OG	0.98		*Rbbp5*	5	−5	OG	0.809
	*Pcdhb18*	5	−4.98	OG	0.97		*Ivl*	5	−5	OG	0.809
	*Sap30bp*	5	−5	OG	0.967		*Nup93*	5	−5	OG	0.809
	*Ube2q2*	5	−5	OG	0.965		*Plscr1*	5	−5	OG	0.809
	*Olfr213*	5	−4.99	OG	0.942		*Olfr380*	5	−5	OG	0.802
	*Trav14-1*	5	−5	OG	0.942		*Ugt1a10*	5	−5	OG	0.802
	*Naip2*	5	−5	OG	0.937		*Arcn1*	2.76	5	TSG	0.932
	*AY358078*	5	−5	OG	0.852		*Vmn2r28*	2.92	5	TSG	0.917
	*Gm14443*	5	−5	OG	0.852		*Calr*	−5	5	TSG	0.91
	*Pdcd10*	5	−5	OG	0.847		*Cd244a*	−2.1	5	TSG	0.91
	*Ss18*	5	−5	OG	0.834		*Ptpdc1*	5	3.54	TSG	0.892
	*Bud31*	5	−5	OG	0.822		*Foxn2*	−5	3.54	TSG	0.892
	*Klra9*	5	−5	OG	0.809		*Ugcg*	−2.1	5	TSG	0.877
	*Cdk8*	5	3.55	TSG	0.859		*Coq2*	0.79	3.41	TSG	0.842
	*Chuk*	2.67	5	TSG	0.859		*Gcsh*	−5	5	TSG	0.842

## Data Availability

No new data were created.

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
