# Peer review of "A Mouse-Specific Model to Detect Genes under Selection in Tumors"

_cancers, 2023, doi:10.3390/cancers15215156_

Round 1
Reviewer 1 Report
This paper addresses an important gap in cancer research by proposing a novel method, GUST-Mouse, for identifying cancer-related genes in mouse models. The paper is well-written, original, and provides clear insights into the adaptation of the GUST model for mouse data and its applications. The use of transfer learning from human exomes is innovative, and the results suggest the potential utility of this approach. Overall, the paper contributes important insights to the field and provides a valuable tool to the scientific community.
I have carefully read the updated version of the paper and the reviewers' comments and believe that the authors perfectly addressed the reviewers' concerns. In my view, this paper can be published after a minor revision.
The aim of this study is very important because the extension of the authors' previous human-based method to the mouse is badly needed. A lot of cancer studies are made on the mouse and the results should be somehow translated to humans. Therefore, one should take into account all similarities and differences between these species. On the other hand, the scarcity of curated murine cancer-related genes hinders the conventional verification of the machine-learning method. The authors suggested a creative approach to circumvent this obstacle and, in my opinion, it is suitable. However, they introduced it only in the Results section ("3.5. Human-mouse comparisons"). I believe that it should be mentioned already in the end of Methods (at least briefly) so that the reader can grasp all the methodology from the beginning and was prepared to this substitute of traditional verification.
As another addition, it would be instructive to mention a paper, which showed that although purifying selection is on average weaker in the primates compared to rodents, it is stronger concentrated on the 'information technology' of life (regulation of gene expression and development), whereas in the rodents, it is biased in favor of metabolism, transport, and energetics (DOI: 10.1016/j.ygeno.2018.02.015). May the comparison of human and mouse cancer-related genes found by the authors be consistent with this pattern? Or the data are insufficient? A bit of discussion on this point would be useful.
Reviewer 2 Report
In the manuscript submitted by Chen and colleagues, they developed the GUST-mouse to estimate long-term and short-term evolutionary selection in mouse tumors, and distinguish between oncogenes, tumor suppressor genes, and passenger genes using high through-put sequencing data. This is an interesting work but needs further revision for accepting for publication.
1. I think a workflow should be provided to reflect the development, validation, and application of your algorithm.
2. Although TCGA dataset was employed to verify the reliability of this model, the heterogeneity between human tumor and mouse tumor should not be neglected. The author should conduct relevant animal study and compare the performance of GUST mouse with classic method to detect the gene mutation status. Only in this way, the reliability of the model could be guaranteed.
3. How about the performance of this model to other mouse models of cancer, such as lung cancer (predict the mutation status of EGFR, ALK, ROS1, MET, KRAS, and etc.)?
4. Different mouse models may undergo distinct intervention and may use different sequencing technology. How could the author guarantee the comparability between these datasets they used in their study? This further support comment 2.
Fine.
Round 2
Reviewer 2 Report
Accept